# Numerical Analysis of Signal Response Characteristic of Piezoelectric Energy Harvesters Embedded in Pavement

**DOI:** 10.3390/ma13122770

**Published:** 2020-06-18

**Authors:** Hailu Yang, Qian Zhao, Xueli Guo, Weidong Zhang, Pengfei Liu, Linbing Wang

**Affiliations:** 1National Center for Materials Service Safety, University of Science and Technology Beijing (USTB), Beijing 100083, China; yanghailu@ustb.edu.cn (H.Y.); zhaoqian928@126.com (Q.Z.); 18291923897@163.com (X.G.); zwd@ustb.edu.cn (W.Z.); 2Research and Development Center of Transport Industry of New Materials, Technologies Application for Highway Construction and Maintenance, Beijing 100088, China; 3Institute of Highway Engineering (ISAC), RWTH Aachen University, 52074 Aachen, Germany; 4Joint USTB Virginia Tech Lab on Multifunctional Materials, USTB, Virginia Tech, Department Civil & Environmental Engineering, Blacksburg, VA 24061, USA

**Keywords:** piezoelectric energy harvester, finite element method, open circuit voltage, moving load

## Abstract

Piezoelectric pavement energy harvesting is a technological approach to transform mechanical energy into electrical energy. When a piezoelectric energy harvester (PEH) is embedded in asphalt pavements or concrete pavements, it is subjected to traffic loads and generates electricity. The wander of the tire load and the positioning of the PEH affect the power generation; however, they were seldom comprehensively investigated until now. In this paper, a numerical study on the influence of embedding depth of the PEH and the horizontal distance between a tire load and the PEH on piezoelectric power generation is presented. The result shows that the relative position between the PEH and the load influences the voltage magnitude, and different modes of stress state change voltage polarity. Two mathematic correlations between the embedding depth, the horizontal distance, and the generated voltage were fitted based on the computational results. This study can be used to estimate the power generation efficiency, and thus offer basic information for further development to improve the practical design of PEHs in an asphalt pavement.

## 1. Introduction

With the continuous development of society, more and more new technologies are applied in road engineering to meet the needs of energy saving, environmental protection, and intelligent infrastructure [1,2,3]. The rapid digitalization of society has greatly promoted the rise and progress of intelligent road transport systems. Sensors are embedded in infrastructure and serve as intelligent nodes in a communication network. These are limited by the requirement of traditional centralized power supply or battery power supply. The development of low-cost, decentralized, and sustainable energy is a necessity to facilitate a wide application of embedded, off-grid sensors [4]. Environmental energy harvesting is the process of transforming the energy that exists in the environment (such as light, heat, mechanical, electromagnetic, biological, and wind energy, among others) into electricity that can be used; it is a potential way to extend the service life of embedded, off-grid sensors [5]. Pavements are designed for millions of axle’s loads during service, which absorbs large amount of wasted mechanical energy. In recent years, researchers have paid more attention to pavement energy harvesting [6,7].

On the basis of the analysis of vehicle-induced vibrations on bridges and pavements, Ashebo et al. addressed the feasibility of vibration energy harvesting from transportation infrastructures to power wireless sensors with a peak power output of 1 mW [8]. It has been found that piezoelectric technology exhibited the highest power density for vibration energy harvesting [9]. The piezoelectric effect is an electromechanical coupling effect, which includes electrical and mechanical boundary conditions [10]. Piezoelectric materials are anisotropic and this property affects their working modes. The working mode of piezoelectric energy harvester (PEH) is determined by its stress state when it is embedded in the pavement. The working modes corresponding to different stress states are shown in Figure 1. In this case, the polarization direction of piezoelectric materials is fixed along the Z axis. When the stress direction is perpendicular to the polarization direction, the working mode is 3-1. When the stress direction is parallel to the polarization direction, the working mode is 3-3. Particularly, at the stress state shown in Figure 1a, the positive charge is generated on the upper surface of the PEH, while the negative one is generated on the lower surface in the 3-1 mode. The sign of the charge will be opposite when the stress state is converse. The relationship between the stress state and the sign of the charge in the 3-3 mode is shown in Figure 1b.

Numerous piezoelectric energy harvesters (PEHs) have been designed and analyzed in different fields [11,12,13,14]. Zhao et al. designed a cymbal-shaped energy harvester for asphalt pavement. Its performance was analyzed for different sizes. The result suggested that the thicker piezoelectric transducers (PZTs) and smaller total cymbal size could increase its efficiency [15]. On the basis of the cymbal structure, Yesner et al. provided a novel electrode design to promote bridge transducer [16]. The exact output energy amount was calculated by Moure et al. considering different kinds of traffic loading by cymbal harvesters [17]. Vázquez-Rodríguez et al. presented a new prospect using lead-free piezoelectric ceramics to determine their behavior in piezoelectric-based road traffic energy harvesting applications [18]. The PEHs of the stack structure have been studied as well [19]. Two numerical methods of the electromechanical model of multilayer PZT stacks were proposed by Zhao and Erturk [20]. Ling et al. designed a kind of piezoelectric bridge transducer that produced a peak voltage of 154 V under 0.7 MPa with 5 Hz half-sine loads [21]. When applied in field experiments and pilot projects, the piezoelectric technology showed potential in pavement energy harvesting, and more analyses on the factors affecting the efficiency have been conducted [22]. Some researchers have figured out that the ambient temperature has a great impact on output power of PEHs and there is a linear relationship between the loading frequency and the output open circuit voltage [23]. Studies have found that the axle load and the axle number of passing vehicles have a significant effect on the power generation of the PEHs [24]. Furthermore, researchers have made numerous finite element (FE) studies on pavement materials and structures with PEHs [25,26,27]. Liu et al. studied the mechanical responses of the asphalt pavement embedded with PEHs and provided the basis for optimization design of PEHs [28].

A review of the literature shows that the wander of the tire load and the embedding location of the PEH in the pavement may significantly affect the power generation of the PEH. However, among the existing simulations and experimental analyses, their influence on the power generation efficiency has not been sufficiently analyzed. Therefore, in this paper, FE pavement models with a moving tire load were developed and verified according to a demonstration pavement embedded with PEHs. On the basis of the numerical model, the influence of different depths of PEH on the output energy was analyzed. The influence of the distance between the load and the PEH on piezoelectric power generation performance was also analyzed. The conclusions can be used to estimate the power generation efficiency of the PEH, and thus offer basic information for further development to improve the practical design of PEHs in asphalt pavement.

## 2. Demonstration Pavement Embedded with PEHs

The pre-fabricated PEHs were embedded in the Highway G320, a highway near the City of Kunming in China. Taking into consideration the contact patch of the tires, it was decided to design the PEH in a square shape with a side length of 0.3 m. The PEH was 0.07 m thick. The PEH was composed of four component parts: the packaging materials, the piezoelectric units, the full bridge rectifiers, and other components for sealing and fastening, as shown in Figure 2. Piezoelectric material was the core component of the PEHs. The piezoelectric unit was a cylindrical structure with a diameter of 20 mm and a height of 23 mm. The type of piezoelectric ceramics was PZT-5H. PA66 nylon with 30% glass fiber was selected for the protective packaging of the PEH owing to its high toughness, load resistance, strength, and resistance to repeated shocks. There were upper, middle, lower layers, and a middle frame, where the upper layer directly undertook the vehicle load, while the ground reaction force was supported by the lower layer, and the middle layer was left to facilitate the holes for nine piezoelectric units and nine full bridge rectifiers.

The output power exited via the cable once the piezoelectric units were connected to the circuit board. Water leakage was prevented by the rectifier bridge being sealed with electronic glue. This was achieved via the application of a silicone gasket between the upper and the lower encapsulation structure. To avoid stress concentration, a 0.03 m diameter stainless steel gasket was inserted between the piezoelectric materials, and then wrapped using a protection package. Therefore, the PEH had a good performance level in terms of compression, resistance to fatigue, and water resistance. Light-emitting diodes (LEDs) were connected to the PEH to receive the piezoelectric power.

Figure 3 shows the PEH installation process and the final state of the demonstration pavement. More information and results about this demonstration pavement can be found in the previous study [29].

## 3. Development and Validation of Finite Element Model

In pavement engineering, full-scale tests are one of the most important methods for material testing and structural analysis. However, the cost of field tests has always been high; furthermore, the conditions cannot always be controlled completely in field tests [30]. Therefore, numerical simulations have been widely applied in research and have been accepted as an effective and efficient substitute [31].

In this study, the simulation and analysis of the signal response characteristic of the PEH embedded in the pavement was conducted with the FE analysis software ANSYS. The FE model was generated based on the aforementioned demonstration pavement structures and the respective material parameters. The pavement structure was simplified as a multi-layer elastic model, which was divided into three layers: surface layer, base layer, and subgrade. The specific parameters are given in Table 1. The model dimensions are as follows: length × width × thickness = 10 m × 6 m × 2.5 m.

The full-scale pavement model is shown in Figure 4, in which the different colors represent the different layers. In ANSYS, the element type of SOLID185 was selected to model the three-dimensional (3D) pavement structures. In order to reduce calculation cost and speed up the simulation, the mesh density gradually becomes sparse from the middle to both sides. The surface mesh density is the largest and the mesh becomes coarser as the depth increases; the total number of elements is 700,000.

In the surface layer, 24 elements with element type of SOLID5 were set to be piezoelectric materials representing a PEH. These elements have a 3D piezoelectric and structural field capability with coupling between the fields. In this research, the PEH was set to be anisotropic and its input parameters include density ρ, dielectric constant ε, stiffness **C**, and piezoelectric stress constant **e.** These parameters were determined based on laboratory test and the values are listed in Table 2. For the sake of brevity, the details about the definition of the material properties can refer to other research [32].

The horizontal position of the PEH was in the center of the pavement. The buried depth of the PEH was defined as the distance from the pavement surface to the upper surface of the PEH. In order to study the influence of the buried depth on the piezoelectric voltage, five pavement models with different buried depths of the PEH were created, and the corresponding depths are 2 cm, 4 cm, 6 cm, 8 cm, and 10 cm, respectively.

The displacements of nodes on the four boundary sides of the model were constrained in the X and Y directions, while the displacements of the bottom surface nodes were constrained in the X, Y, and Z directions. The other nodes were not constrained. Because of the existence of electrodes, the upper and lower surfaces of the PEH were considered as voltage equipotential surfaces. In the FE model, the electrode was simulated by coupling the voltage degrees of freedom of the nodes on the upper and lower surfaces, respectively. The nodes coupled with voltage degrees of freedom would keep the same voltage value in the simulation process. Moreover, the voltage of the bottom surface of the PEH was constrained to 0 V.

A truck tire load was applied in the model, which was assumed as rectangular with 30 cm wide and 20 cm long, as shown in Figure 4. It moved along the longitudinal direction with a contact pressure of 0.7 MPa. Five different loading paths were considered in the simulation to investigate the influence of the tire load on the piezoelectric voltage, that is, the horizontal distances between the center of the load and the center of the PEH were set as 0, 0.2, 0.4, 0.6, and 0.8 m, respectively. The moving speed of the tire was equivalent to a velocity of 40 km/h, which was determined by setting up a step function in a different duration on any of the passing elements [33]. During the loading process, the upper surface voltage of the piezoelectric structures was monitored.

The numerical model was verified by comparing the computational value with measured ones derived from the demonstration pavement. In the comparison, the tire load directly passes the PEH and the buried depth of the PEH is 2 cm. The peak voltage derived from the simulation is 626 V, whereas those obtained from the two measurements are 704 V and 608 V, respectively. A range of error of 20% is considered to allow for uncertainties and fluctuations. As the absolute values of relative error are 11% and 3% between the simulation and measurement, based on this criterion, the reliability of the developed FE model is validated.

## 4. Results and Discussion

### 4.1. Open Circuit Voltage Response of PEH

The FE simulations were carried out based on the validated models. Different horizontal distances between the tire load and the PEH as well as the different buried depths of the PEH were considered. The numerical piezoelectric voltages of the open circuit structure are shown in Figure 5.

In general, the responses of the open circuit voltage of PEH at different buried depths are similar, that is, the changes over time are similar, while the values are different. Particularly, when the tire load passes directly above the PEH (the horizontal distance equals to 0 m), the stress state of the PEH is first 3-1 mode, then 3-3 mode, and finally 3-1 mode. The output from the 3-1 mode and 3-3 mode produces positive and negative voltage, respectively. Therefore, the values of the open circuit voltage are first positive, then become negative, and finally return to be positive. When the loading path does not pass directly above the PEH, it is not subjected to vertical compression, the main stress is horizontal stress, and the stress state is shifted from 3-3 mode to 3-1 mode. The power generation is a combination of the two modes. By comparison, 3-3 mode has high power generation capacity, but its corresponding range is small, while 3-1 mode has the opposite effect. As a result, in the application of pavement energy harvesting, the design of embedding location of the PEH should consider the spatial distribution of the traffic vehicle load.

### 4.2. Maximum Positive Voltage of PEH

The positive voltage peaks at different buried depths of the PEH and horizontal distances are concluded and listed in Table 3. In order to see the changes, the data are drawn in Figure 6. 

It can be seen that, when the load passes directly above the PEH, the peak voltage decreases sharply with the increase of the buried depth. When the horizontal distance between the load and the PEH is 20 cm, the buried depth of the PEH still has a great influence on the response, that is, the piezoelectric voltage decreases significantly as the depth increases. When the horizontal distance reaches and exceeds 40 cm, the influence of the buried depth of the PEH on the open circuit voltage is sharply reduced. It is worth noting that the peak voltage produced by the PEH with horizontal loading distance of 20 cm is larger than the one in case the load passes directly above the PEH. Complex stress state within the pavement structure is generated when the pavement bears the moving tire load. As the PEH is embedded in the pavement, it withstands not only the vertical stress, but also the horizontal stresses. The voltage polarity produced by the vertical stress is opposite to the polarity of the voltage produced by the horizontal stresses. Therefore, the piezoelectric output is not necessarily better the closer it is to the load, as it depends on its stress state.

To comprehensively quantify the correlation among horizontal distance, buried depth, and peak voltage, the data were fitted with the aid of the software TableCurve 3D. A 3D curve fitting plot is shown in Figure 7. The influence of the horizontal distance (D) and buried depth (H) on the peak voltage (U) is made apparent in the 3D curve fitting image. The fitting equation is shown below:(1)$$\begin{array}{c} U=159.53+3.19D+0.88H-2.67D^{2}-0.11H^{2}+0.34\mathrm{DH}+0.12D^{3}\\ +0.0010H^{3}-0.0032{\mathrm{DH}}^{2}+0.0075D^{2}H \end{array}$$

The goodness-of-fit of the fitting equation is plotted in Figure 8. The correlation coefficient R^2^ of the equation is 0.95, which is high enough to prove the good applicability of the fitting equation. With this equation, the peak voltage produced by the PEH can be estimated based on the loading distance and buried depth of the PEH, and thus promote optimization of the pavement design with PEHs.

### 4.3. Maximum Negative Voltage of PEH

As aforementioned, when the tire load passes directly above the PEH (the horizontal distance equals to 0 m), the stress state of the PEH is first 3-1 mode, then 3-3 mode, and finally 3-1 mode. The output from the 3-3 mode produces negative voltage. The peak negative voltages derived from this process are presented in Figure 8. As can be seen from Figure 9, the absolute value of the voltage produced by the PEH increases first and then decreases with the increase of the buried depth of the PEH. The reason can be that the polarity and magnitude of the voltage are the result of the entire stress state born within the PEH. The stress evolution along the pavement depth does not change linearly, which has been proven in previous research [34]. The maximum open circuit voltage is the highest when the PEH is buried at the depth of 6 cm and has an output of −779.2 V. The polynomial fitting equation for the relationship between open circuit peak voltage (U) and buried depth (H) is as follows:(2)$$U=-0.3846H^{3}+13.215H^{2}-125.38H-424.99$$

The correlation coefficient R^2^ of the equation is 1, which can prove the good applicability of the fitting equation. According to Equation (2), the buried depth H corresponding to the maximum open circuit voltage is 6.7 cm, and the maximum open circuit voltage is −787.49V.

## 5. Conclusions and Future Work

Owing to the change of stress distributions in the pavement under moving load, the stress state of PEH is not simple, which in turn affects the piezoelectric response. A full scale pavement model with a moving tire load was developed based on the FE method. The influence of buried depth of the PEH and the horizontal distance between the tire load and the PEH on piezoelectric power generation was investigated. The following results can be concluded:

(1) The responses of the open circuit voltage of PEH at different buried depths are similar. When the tire load passes directly above the PEH, the values of the open circuit voltage are first positive, then become negativem and finally return to being positive. When the loading path does not pass directly above the PEH, the value of the open circuit voltage is always positive.

(2) When the load passes directly above the PEH, the peak voltage decreases sharply with the increase of the buried depth. When the horizontal distance between the load and the PEH is 20 cm, the buried depth of the PEH still has a great influence on the response. When the horizontal distance reaches and exceeds 40 cm, the influence of the buried depth of the PEH on the open circuit voltage is sharply reduced.

(3) The power generation is a combination of the 3-1 mode and 3-3 mode. By comparison, 3-3 mode has high power generation capacity, but its corresponding range is small, while 3-1 mode has the opposite effect. The piezoelectric output is not necessarily better the closer it is to the load; it depends on its stress state.

(4) Two mathematic correlations between the buried depth, the horizontal distance, and the generated voltage were fitted based on the computational results. The correlation coefficients of the equations are high enough to prove their good applicability.

The result derived from this study is feasible to be used to estimate the power generation efficiency of the PEHs under different working situations, and thus provide an opportunity to improve the practical design of the pavement with the PEHs. For the future work, a series of PEH with different arrangement methods will be considered. Viscoelasticity of the asphalt materials will be taken into account to compute the recoverable and irrecoverable deformations of asphalt pavements with PEH [35]. A real tire-pavement interaction FE model will be created by simulating an explicit passing tire [36]. The influence of the temperature on the mechanical responses of the asphalt pavement with PEH will be studied as well.

## Figures and Tables

**Figure 1 materials-13-02770-f001:**
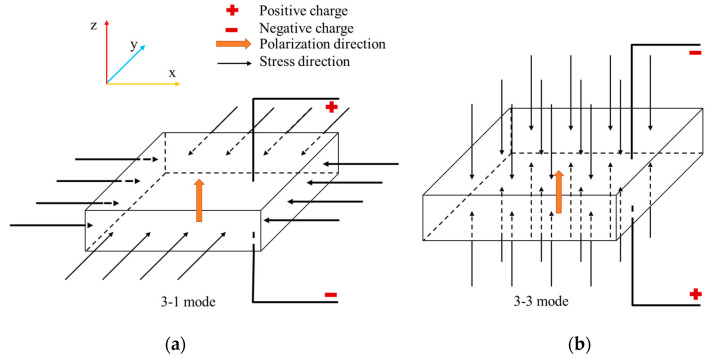
Working modes of the piezoelectric energy harvester (PEH): (**a**) 3-1 mode; (**b**) 3-3 mode.

**Figure 2 materials-13-02770-f002:**
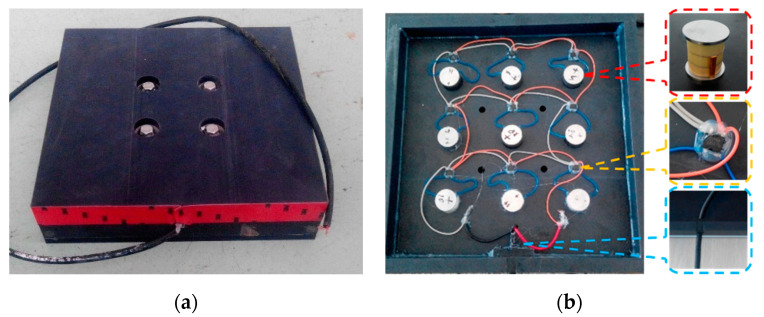
Illustration of the PEH: (**a**) outward appearance; (**b**) internal structure.

**Figure 3 materials-13-02770-f003:**
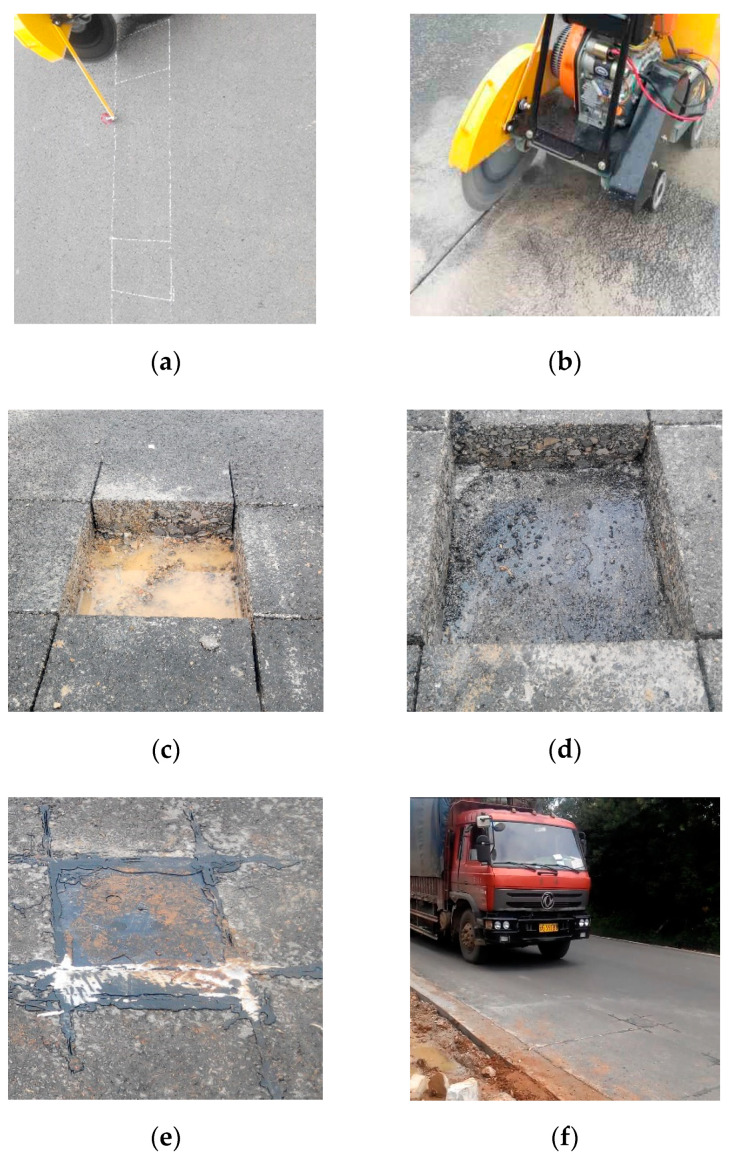
The flow chart of PEH installation: (**a**) positioning and scribing; (**b**) slotting; (**c**) coring for installation; (**d**) cleaning and levering the holes’ bottom; (**e**) putting PEHs in the pavement and laying the cable; (**f**) final state of the demonstration pavement.

**Figure 4 materials-13-02770-f004:**
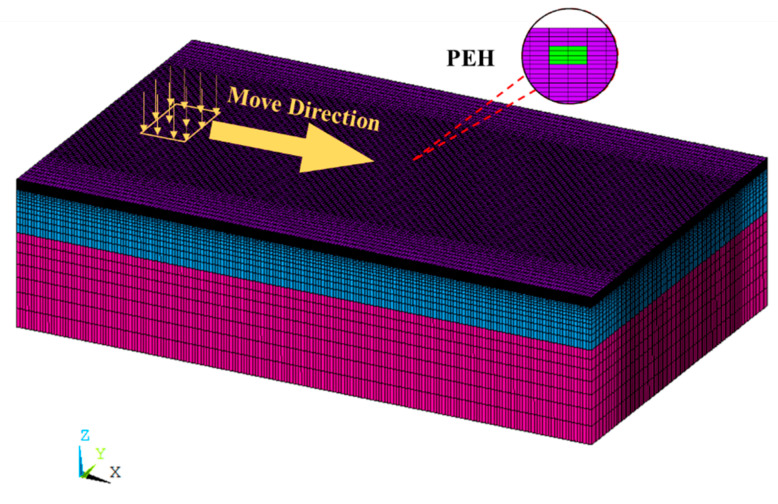
Full scale pavement model with ANSYS.

**Figure 5 materials-13-02770-f005:**
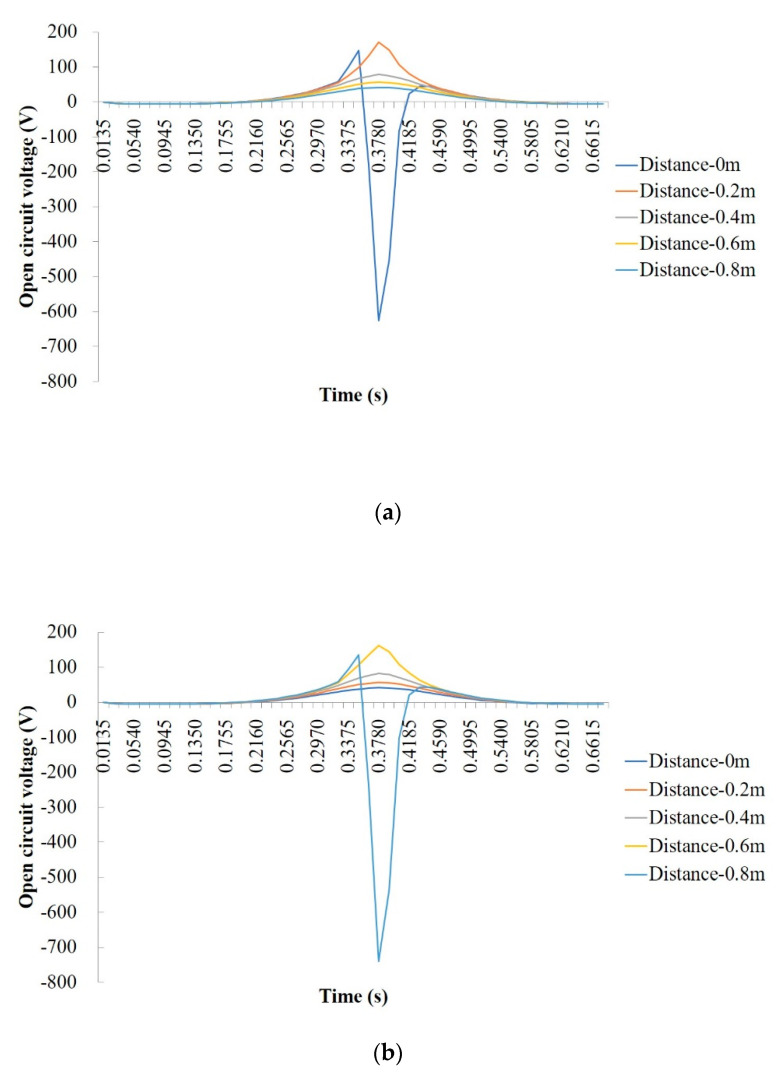
Open circuit voltage response of PEH at different horizontal loading distances and buried depths: (**a**) buried depth of 2 cm; (**b**) buried depth of 4 cm; (**c**) buried depth of 6 cm; (**d**) buried depth of 8 cm; (**e**) buried depth of 10 cm.

**Figure 6 materials-13-02770-f006:**
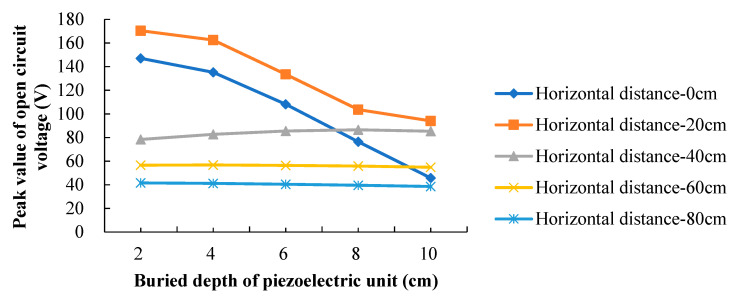
Response of PEH at different buried depths with different horizontal loading distances.

**Figure 7 materials-13-02770-f007:**
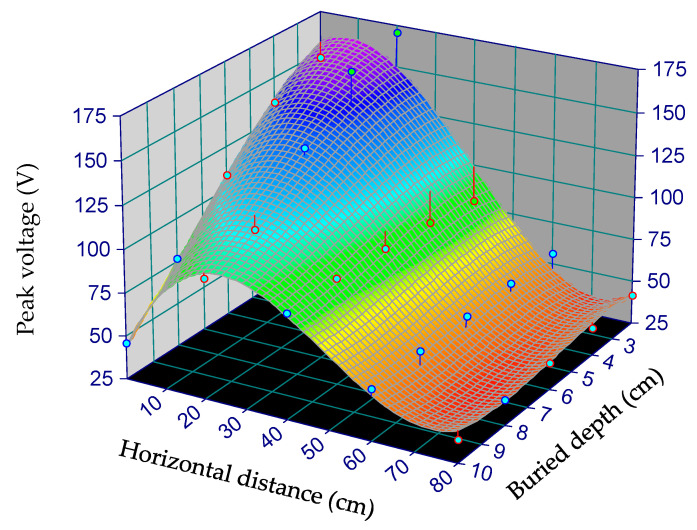
The 3D curve fitting image of horizontal distance D, buried depth H, and peak voltage U.

**Figure 8 materials-13-02770-f008:**
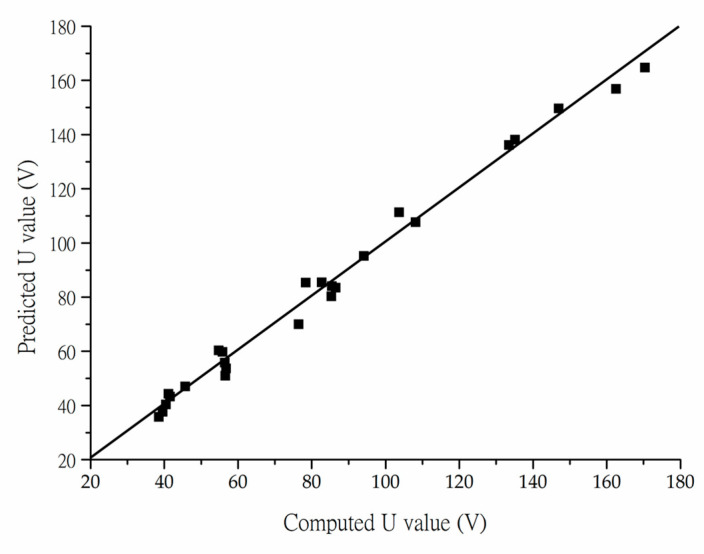
The goodness-of-fit plot for the fitting equation.

**Figure 9 materials-13-02770-f009:**
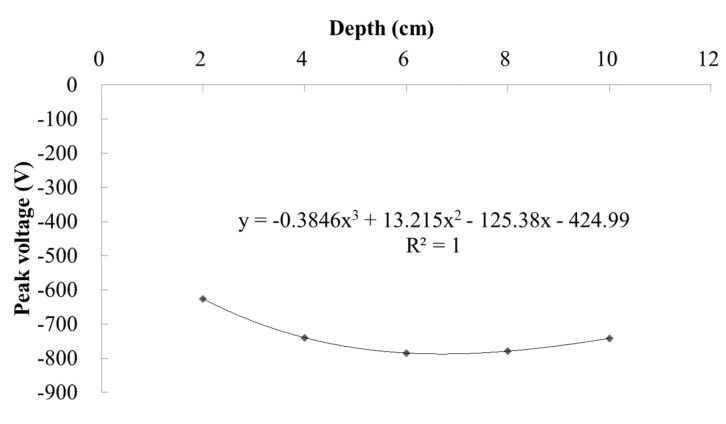
Curves of maximum negative voltages with buried depth when load passes directly above the PEH.

**Table 1 materials-13-02770-t001:** Geometry and material parameters of the pavement model.

Layer	Material	Thickness/m	Young’s Modulus/MPa	Poisson’s Ratio	Density/kg·m^−3^	Damping Coefficient α
Surface	AC-20	0.18	1300	0.35	2400	0.25
Base	cement stabilized macadam	0.72	1600	0.30	2100	0.25
Subgrade	Soil	1.60	50	0.40	1900	0.25

**Table 2 materials-13-02770-t002:** Material properties for the piezoelectric energy harvester (PEH).

$\mathrm{Density}\rho \;(\mathrm{kg}\cdot m{-}^{3})$	Dielectric Constant ε (C/Vm)	Stiffness C (MPa)	Piezoelectric Stress Constant e (C/m^2^)
7500	**ε**_11_ = $3.27\times 10^{-9}$	**C**_11_ = $13.9\times 10^{10}$	**e**_31_ = −5.2
-	**ε**_33_ = $5.16\times 10^{-9}$	**C**_12_ = $7.78\times 10^{10}$	**e**_33_ = 15.1
-	-	**C**_13_ = $7.43\times 10^{10}$	**e**_15_ = 12.7
-	**-**	**C**_33_ = $11.5\times 10^{10}$	-
-	**-**	**C**_44_ = $2.56\times 10^{10}$	-
-	**-**	**C**_66_ = $3.06\times 10^{10}$	-

**Table 3 materials-13-02770-t003:** Maximum positive voltage of PEH.

Buried Depth	Peak Voltage at Different Horizontal Distances (V)				
	0 cm	20 cm	40 cm	60 cm	80 cm
2 cm	146.99	170.40	78.36	56.58	41.56
4 cm	135.15	162.534	82.73	56.76	41.18
6 cm	108.14	133.49	85.50	56.44	40.46
8 cm	76.44	103.68	86.51	55.82	39.60
10 cm	45.72	94.08	85.36	54.80	38.54

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
