# Peer review of "Numerical Analysis of Signal Response Characteristic of Piezoelectric Energy Harvesters Embedded in Pavement"

_materials, 2020, doi:10.3390/ma13122770_

Round 1
Reviewer 1 Report
Thank you for your investigation.
- In my opinion, Figure 4. should be correct. Time range from 0.0135 to 0.2565 and from 0.4995 to 0.6615 is not informative.
- I did not find the boundary condition for simulation from the text. Fixed constraints (I guess bottom part of the pavement), maybe a slider condition from the sides, or infinite domain from pavement's sides or similar.
- You are concluded: 3) The power generation is a combination of the 3-1 mode and 3-3 mode. By comparison, 3-3 mode has high power generation capacity, but its corresponding range is small, while 3-1 mode has the opposite effect... > polarization of piezoceramic, which you used, by the thickness (d33). According to the experiments, the main pavement deformation close to the bending mode, I think piezoceramic orientation inside the pavement should have an influence on the EH efficiency, isn't it?!
Reviewer 2 Report
The novelity of presented results with earlier developments of authors must be clerfied.
The finite element matematical model is upsent. Presenting of values of specifed matrices without formula is not correct.
Working modes of pjezoelectric materials are not clearly described. Using of notations 3.1, 3-3 needs to be motyvated motivation.e not clearly described.
Comments on the future work, dont belong to conclusions.
Author Response
Thanks for your comments. All comments have been carefully reviewed and addressed in the revised paper. The response to the comments has been uploaded as a Word file.
Please see the attachment

Reviewer 3 Report
The paper is well structured and written. The conclusions are supported by the analysis of the simulation data presented and therefore the paper can be accepted for publications as it is. I only suggest to revise a couple of sentences that sound excessive to me. I will send a pdf file with few suggestions for the authors.

Author Response

(The authors gave the same response as above.)
